# The African Swine Fever Virus with MGF360 and MGF505 Deleted Reduces the Apoptosis of Porcine Alveolar Macrophages by Inhibiting the NF-κB Signaling Pathway and Interleukin-1β

**DOI:** 10.3390/vaccines9111371

**Published:** 2021-11-22

**Authors:** Qi Gao, Yunlong Yang, Weipeng Quan, Jiachen Zheng, Yizhuo Luo, Heng Wang, Xiongnan Chen, Zhao Huang, Xiaojun Chen, Runda Xu, Guihong Zhang, Lang Gong

**Affiliations:** 1Key Laboratory of Zoonosis Prevention and Control of Guangdong Province, College of Veterinary Medicine, South China Agricultural University, Guangzhou 510462, China; qigao2021@scau.edu.cn (Q.G.); yunlongyang@stu.scau.edu.cn (Y.Y.); weipengquan@stu.scau.edu.cn (W.Q.); zhengjc@stu.scau.edu.cn (J.Z.); lawzz@stu.scau.edu.cn (Y.L.); wangheng2009@scau.edu.cn (H.W.); cxn201314@stu.scau.edu.cn (X.C.); yingwenmulu@stu.scau.edu.cn (Z.H.); xiaojunchen@stu.scau.edu.cn (X.C.); 18819255305@163.com (R.X.); 2Maoming Branch, Guangdong Laboratory for Lingnan Modern Agriculture, Maoming 525000, China; 3Research Center for African Swine Fever Prevention and Control, South China Agricultural University, Guangzhou 510642, China; 4African Swine Fever Regional Laboratory of China (Guangzhou), Guangzhou 510642, China

**Keywords:** African swine fever virus, MGF, CD2v, NF-κB, IL-1β, apoptosis

## Abstract

African swine fever virus (ASFV) poses serious threats to the swine industry. The mortality rate of African swine fever (ASF) is 100%, and there is no effective vaccine currently available. Complex immune escape strategies of ASFV are crucial factors affecting immune prevention and vaccine development. CD2v and MGF360-505R genes have been implicated in the modulation of the immune response. The molecular mechanisms contributing to innate immunity are poorly understood. In this study, we discover the cytopathic effect and apoptosis of ΔCD2v/ΔMGF360-505R-ASFV after infection in porcine alveolar macrophages (PAMs) was significantly less than wild-type ASFV. We demonstrated that CD2v- and MGF360-505R-deficient ASFV decrease the level of apoptosis by inhibiting the NF-κB signaling pathway and IL-1β mRNA transcription. Compared with wild-type ASFV infection, the levels of phospho-NF-κB p65 and p-IκB protein decreased in CD2v- and MGF360-505R-deficient ASFV. Moreover, CD2v- and MGF360-505R-deficient ASFV induced less IL-1β production than wild-type ASFV and was attenuated in replication compared with wild-type ASFV. We further found that MGF360-12L, MGF360-13L, and MGF-505-2R suppress the promoter activity of NF-κB by reporter assays, and CD2v activates the NF-κB signaling pathway. These findings suggested that CD2v- and MGF360-505R-deficient ASFV could reduce the level of ASFV p30 and the apoptosis of PAMs by inhibiting the NF-κB signaling pathway and IL-1β mRNA transcription, which might reveal a novel strategy for ASFV to maintain the replication of the virus in the host.

## 1. Introduction

African swine fever (ASF) is a highly pathogenic animal infectious disease caused by the African swine fever virus (ASFV) infecting domestic pigs and wild boars, causing huge economic losses to the pig industry [1]. ASFV was first reported in Kenya in 1921, and the disease is spreading in Africa, Europe, and Asia, posing a serious threat to the global economy and ecology [2,3]. The lethality rate of ASFV high-viral strain infection is as high as 100%, while the low-viral strain can cause latent infection of the host, and animals can therefore develop immunity against this strain [4]. However, there is currently no effective ASFV vaccine for prevention and control, and no drugs have been approved to treat ASF [5]. ASFV infection can evade the host’s defense system and thus escape the host’s innate antiviral immunity to ASFV [6]. The huge genome and complex immune escape mechanism of ASFV pose a huge challenge to the development of high-efficiency ASFV vaccine [7].

ASFV is a DNA arbovirus and the only member of the family Asfarviridae [8]. ASFV is an icosahedral virus, encoding a linear dsDNA genome of 165 genes, with a genome length of 170–190 kb [9]. Previous studies have shown that ASFV protein can inhibit the host’s defense mechanism, which is of great significance for the immune escape built by the virus and the maintenance of virus replication [10]. Studies have shown that deleting virulence genes related to virus virulence from the ASFV genome can be used to develop ASFV live attenuated vaccines [1]. The CD2v protein of ASFV plays an important role in the spread of the virus and the immune escape mechanism. It can induce the infected cells to adsorb pig red blood cells in a rosetting of erythrocytes (haemadsorbtion or HAD) [11]. It is also a protein necessary for the adsorption of virus particles and pig red blood cells [12,13]. The pCD2v/pEP402R protein is the only protein that clearly exists on the outer envelope of virus particles, and it has been found to play an important role in ASFV invading cells or spreading between cells [14]. In a study, it was found that deletion of the EP402R gene encoding the CD2v protein of ASFV delayed the appearance of clinical symptoms and reduced the spread of the virus, but did not reduce the mortality rate of viral infections [15]. A recent study showed that deleting the EP402R gene of the BA71V virulent strain will weaken the virulence of the virulent strain and produce protection against the virulent strain [16]. There are five multi-gene families in ASFV, including MGF-100, MGF-110, MGF-300, MGF-360, and MGF-505. They can regulate the host’s immune response mechanism and have host specificity [17,18], however, the functions and mechanisms of these multi-gene families have not yet been thoroughly studied. Among them, the two families of MGF360 and MGF505 exist in the highly variable left-end genomic region and encode products with structural similarity [17,19]. Therefore, the study of MGF360 and MGF505 regions may clarify their role in ASFV replication and their contribution to virus virulence. Studies have shown that deleting the MGF gene in the BA71V strain adapted to Vero cells will reduce the virus replication in macrophages [20,21]. In addition, studies have reported that the MGF360 and MGF505 genes are essential for the virus replication in ticks [22]. There are also reports that the genes in the MGF360 and MGF505 regions have the function of inhibiting host interferon (IFN) production [9]. MGF360 and MGF505 are related to the host specificity, innate immune mechanism, and virulence of ASFV; deletion of MGF will reduce the virulence of ASFV [1].

Nuclear factor-kappa B (NF-kappa B) is a family of transcription factors that can regulate many genes involved in immunity, inflammation, and apoptosis [23]. For a long time, the NF-κB pathway has been regarded as a typical pro-inflammatory signal transduction pathway. Pro-inflammatory factors such as interleukin 1 (IL-1) and tumor necrosis factor α (TNF-α) have an activating effect on NF-κB. The activation of the NF-κB signaling pathway can promote the secretion of inflammatory factors, chemokines, and adhesion molecules [24]. However, inflammation is a complex physiological process, and the inflammatory response is characterized by coordinating and regulating the secretion of pro-inflammatory and anti-inflammatory factors and the activity of various signal transduction pathways [23,25]. NF-κB activation is also widely involved in inflammatory diseases, and many studies have focused on the development of anti-inflammatory drugs for NF-κB [26]. As generally recognized, NF-kappaB exerts an anti-apoptotic action, promoting the survival of defective cells, which may result in the development of several viruses. Nevertheless, recent reports also point to a pro-apoptotic activity of NF-kappaB [27]. A recent study showed that HO-1 can inhibit IL-1β-induced host cell apoptosis by inhibiting the activity of NF-κB [28].

Here we report the construction of a recombinant virus ΔCD2v/ΔMGF360-505R-ASFV derived from the highly virulent GZ201801-ASFV by specifically deleting the genes CD2v and MGF360-505R: MGF505-1R, MGF360-12L, MGF360-13L, MGF360-14L, MGF505-2R, and MGF505-3R. ASFV-ΔCD2v/ΔMGF360-505R replicates efficiently in primary swine macrophage cell cultures lower than the parental virus. The cytopathic effect of ΔCD2v/ΔMGF360-505R-ASFV after infection with PAMs was significantly less than GZ201801-ASFV. Co-infection withΔCD2v/ΔMGF360-505R-ASFV and GZ201801-ASFV can reduce the cytopathic effect of PAMs cells. ΔCD2v/ΔMGF360-505R-ASFV can down-regulate phosphorylation levels of NF-κB and IκB, and decrease the level of IL-1β expression. In reporter assays, expression of MGF360-12L, MGF360-13L, and MGF505-2R inhibited the NF-κB promoter. Found in the Western blot experiment, CD2v activates the NF-κB signaling pathway. Detection by flow cytometry showed that ΔCD2v/ΔMGF360-505R-ASFV infection induced apoptosis of PAMs significantly less than GZ201801-ASFV. Our study explored the relations between the NF-κB signaling pathway and apoptosis in ASFV infection, which may help us better understand the pathogenic mechanism of ASFV.

## 2. Materials and Methods

### 2.1. Cell Culture and Virus

Primary porcine alveolar macrophages (PAMs) were collected from 20–30 day-old SPF (Specific Pathogen Free, SPF) pigs. The study is meaningless in the case of co-infections in the porcine respiratory tract; it will affect the study of the interaction between a single pathogen and the host [29]. Therefore, we found that ASFV, PRV, CSFV, SIV, PCV2, and PRRSV pathogens and antibodies were negative through the pathogenic detection of PAMs. CD163 and CD169 are proteins specifically expressed on the surface of macrophages [30,31]. Therefore, the expression of CD163 and CD169 proteins on the cell surface were detected by flow cytometry, and the result shows that the purity of PAMs is approximately 98% (Appendix A). The high virulence, hemadsorbing ASFV isolate GZ201801 (GenBank: MT496893.1) was isolated in Guangzhou, China, and is p72 genotype Ⅱ. GZ201801-ASFV, ΔCD2v/ΔMGF360-505R-ASFV, ΔCD2v-ASFV, and ΔMGF360-505R-ASFV are preserved in the Infectious Diseases Laboratory of South China Agricultural University. Viruses were inoculated in PAMs, cultured in RPMI 1640 (Gibco, Waltham, MA, USA), and supplemented with 10% fetal bovine serum (FBS; Gibco) in a 37 °C incubator with 5% CO_2_. PK-15 cells were maintained in Dulbecco’s Modified Eagle’s Medium (DMEM; Thermo Fisher Scientific, Waltham, MA, USA) supplemented with 10% fetal bovine serum (FBS; Gibco). Cells were grown at 37 °C in a 5% CO_2_ atmosphere saturated with water vapor in a culture medium supplemented with 2 mM l-glutamine, 100 U/mL gentamycin, and 0.4 mM nonessential amino acids. For all of the cells used, the mycoplasma contamination was checked and excluded by a Mycoplasma stain detection kit (Beyotime, Shanghai, China).

### 2.2. Plasmids

The plasmid NF-κB used for the NF-κB reporter assay was a gift from Professor Haixue Zheng (Lanzhou Veterinary Research Institute, Chinese Academy of Auricula Sciences). The plasmids pCAGGs-MGF360-12L, pCAGGs-MGF360-13L, pCAGGs-MGF360-14L, pCAGGs-MGF505-1R, pCAGGs-MGF505-2R, and pCAGGs-MGF505-3R were constructed using PCR and pEASY-Basic Seamless Cloning and Assembly Kit (Transgen, Beijing, China, CU201-02). All of these plasmids were confirmed by Sanger sequencing and successfully expressed in PK-15 cells (Appendix A).

### 2.3. Reagents and Antibodies

The reagents and antibodies used in this study are α-Tubulin Rabbit Polyclinal Antibody (Beyotime, Shanghai, China, AF0001); rabbit monoclonal antibodies against phospho-NF-κB p65 (Ser536) (93H1) (Cell Signaling Technology, Danvers, MA, USA, 3033T); rabbit monoclonal antibodies against phospho-IκBα (Ser32) (14D4) (Cell Signaling Technology, Danvers, MA, USA, 2859T); IRDye 800CW Goat (polyclonal) Anti-Rabbit IgG (H + L), Highly Cross Adsorbed (LI-COR, 926-32211); IRDye 800CW Goat (polyclonal) Anti-Mouse IgG (H + L), Highly Cross Adsorbed (LI-COR, 926-32210); Dual-Luciferase Reporter Assay System, Promega, USA, E1910; Annexin V-APC apoptosis detection kit (KeyGEN BioTECH, Nanjing, China, KGA1021).

### 2.4. HAD Assay

As previously described in [32], primary PAMs were cultured in 96-well plates and infected with 10-fold diluted ASFV. The HAD_50_ was calculated by using the Reed and Muench method [33]. Primary PAMs were infected with GZ201801-ASFV, ΔMGF360-505R-ASFV, and ΔCD2v/ΔMGF360-505R-ASFV at an MOI of 0.1. HAD assay of the ASFV. The ASFV was inoculated into pig PAMs with 1% pig blood cells. Cultures were observed for HAD phenomena over 7 days.

### 2.5. Polymerase Chain Reaction

DNA was isolated by Axyprep Body Fluid Viral DNA/RNA Miniprep Kit (Axygen, Union City, CA, USA, SV US, AP-MN-BF-VNA). A total of 1 μL of DNA was used for real-time PCR assay using AceQ Universal U+ Probe Master Mix V2 (Vazyme, Nanjing, China, Q513-02). This assay targets the VP72 gene of ASFV [34], and the relative quantity of viral DNA was determined by the CADC p72 primers and probe experiment. Gene-specific primer and probe sequences are listed in Appendix A. Total RNA was isolated by RNAiso Plus (Takara, 9108, Beijing, China). RNA was reverse transcribed into cDNA using the HiScript II 1st Strand cDNA Synthesis Kit ( + gDNA wiper) (Vazyme, China, R212-02). The cDNA was used for real-time PCR assay using ChamQ Universal SYBR qPCR Master Mix (Vazyme, China, Q711-02) kit. The relative quantity of cell RNA was determined by performing a comparative Ct (ΔΔCt) experiment using GAPDH as an endogenous control [35,36,37]. ΔΔCT method assumes 100% efficiency of qPCR assays. Gene-specific primers sequences were designed using Oligo7 software, listed in Table 1.

### 2.6. Western Blot Analysis

For the Western blot analysis, cells were lysed in RIPA buffer (Beyotime, Shanghai, China) and denatured by adding 4× Laemmle SDS-PAGE buffer (containing DL-Dithiothreitol), followed by heating for 15 min at 100 °C. The proteins were then separated on SDS-PAGE gels and transferred onto NC membranes by a Trans-Blot Turbo rapid transfer system (Bio-Rad, Hercules, CA, USA) according to the manufacturer’s instructions. The membranes were blocked in 5% defatted milk (dissolved in Tris-buffered saline (TBS)) for 1 h at 37 °C and then incubated with a primary antibody for 1 h at room temperature or overnight at 4 °C. The membranes were then washed extensively in wash buffer (TBS containing 0.1% Tween 20) three times (for 5 min each time) with agitation and incubated with an IRDye 800CW secondary antibody for 1 h at 37 °C according to the species source of the primary antibody. The membranes were washed three times in wash buffer and imaged using an Odyssey (LICOR, Lincoln, NE, USA) to visualize the protein bands. α-Tubulin was utilized as a loading control.

### 2.7. Transfection and Luciferase Reporter Assays

Cells were transfected using the Lipofectamine 2000 transfection reagent (Thermo Fisher Scientific, Waltham, MA, USA). Due to cell characteristics, the plasmid transfection efficiency of PK-15 cells is 40–50%. For the NF-κB reporter assay, NF-κB-Luc reporter plasmid and luciferase gene internal reference plasmid pRL-TK were mixed and co-transfected with the control empty vector, pCAGGs-MGF360-12L, pCAGGs-MGF360-13L, pCAGGs-MGF360-14L, pCAGGs-MGF505-1R, pCAGGs-MGF505-2R, and pCAGGs-MGF505-3R, respectively. At 6 h post transfection, culture supernatants were replaced with DMEM without FBS. At 24 h post transfection, cells were collected and subjected to further analysis. Cells were lysed and measured using a dual luciferase assay kit (Promega, Madison, WI, USA) according to the manufacturer’s instructions.

### 2.8. Flow Cytometry

After the cells were digested into single cells with 0.25% trypsin, the cells were washed three times with PBS, and the cells were stained with Annexin V-APC Apoptosis Detection Kit. Use the excitation wavelength of 633 nm and the maximum emission wavelength of 660 nm of the Beckman flow cytometer to measure the cell fluorescence. Analysis was carried out on FlowJo software.

### 2.9. Statistical Analysis

All data of each assay represents at least two separate experiments and were determined in triplicate. The results collected from triplicate determinations were analyzed as the means ± standard deviations (SD). The data was tested by using the Kolmogorov–Smirnov test (with the Dallal–Wilkinson–Lilliefor *p* value) method in GraphPad, which showed that these data are normally distributed. Data difference of each experiment was analyzed by One-Way Analysis of Variance (ANOVA) followed by the Tukey’s *t*-test in GraphPad Prism 8.0 software (San Diego, CA, USA). * *p* < 0.05, ** *p* < 0.01 and *** *p* < 0.001 were considered to be statistically significant at different levels.

## 3. Results

### 3.1. Replication of ΔCD2v/ΔMGF360-505R-ASFV in Primary Swine Macrophages

MGF genes located in the left variable region of the ASFV genome and CD2v gene have been described to be involved in ASFV replication in swine macrophages [20,21,38,39]. The in vitro growth characteristics of ΔCD2v/ΔMGF360-505R-ASFV were evaluated in primary swine macrophage cell cultures and the primary cell targeted by ASFV during infection in swine, and compared to those of the parental GZ201801-ASFV strain in a multistep growth curve analysis. GZ201801-ASFV has HAD phenomenon, but ΔCD2v/ΔMGF360-505R-ASFV does not (Figure 1A. Cell cultures were infected with these viruses at an MOI of 0.1, and samples were collected at 0, 3, 6, and 12 h post-infection (hpi) (Figure 1B). ΔCD2v/ΔMGF360-505R-ASFV displayed a growth kinetic difference from that of the parental GZ201801-ASFV virus. CD2v and MGF360-505R knockout inhibit ASFV replication. The ASFV-p30 protein levels in ΔCD2v/ΔMGF360-505R-ASFV were significantly lower than in the GZ201801-ASFV group (Figure 1C). And the gray analysis of protein bands was carried out (Appendix A). Therefore, deletion of MGF360 and MGF505 genes in ΔCD2v/ΔMGF360-505R-ASFV does significantly affect the ability of the virus to replicate in primary swine macrophage cultures.

### 3.2. Cytopathic Effect of ΔCD2v/ΔMGF360-505R-ASFV in Primary Swine Macrophages

ASFV has a restricted cellular tropism. The primary target cells for replication are macrophages and monocytes, although replication in dendritic cells has also been reported [10,40,41]. Macrophages and monocytes infected with ASFV will produce cytopathic effects, such as vacuolation and shedding. Compared with the control group, the PAMs cells infected with GZ201801-ASFV produced more severe CPE, while the CPE produced by ΔCD2v/ΔMGF360-505R-ASFV-infected PAMs was significantly reduced. When GZ201801-ASFV and ΔCD2v/ΔMGF360-505R-ASFV are co-infected, compared with GZ201801-ASFV-infected PAMs, CPE is significantly reduced (Figure 2), indicating that ΔCD2v/ΔMGF360-505R-ASFV can reduce the CPE caused by GZ201801-ASFV-infected PAMs.

### 3.3. Apoptosis of ΔCD2v/ΔMGF360-505R-ASFV in Primary Swine Macrophages

Apoptosis represents an important innate cellular mechanism to prevent virus infection, and many viruses have developed strategies in turn, for inhibiting or delaying this cellular response [42]. ASFV-infected cells undergo apoptosis [43] and show the characteristic morphological changes of programmed cell death, including typical membrane blebbing of the infected cell that led to the formation of numerous vesicles containing virus [44] and this could be an efficient system for virus spread. After infection with GZ201801-ASFV, 26.49% of PAMs were apoptotic, and after infection with CD2v/ΔMGF360-505R-ASFV, 8.72% of PAMs were apoptotic. When GZ201801-ASFV and CD2v/ΔMGF360-505R-ASFV were co-infected, 9.35% of PAMs were apoptotic. After infection with ΔMGF360-505R-ASFV, 27.67% of PAMs were apoptotic, and after infection with ΔCD2v-ASFV, 30.45% of PAMs were apoptotic. When GZ201801-ASFV and ΔMGF360-505R-ASFV were co-infected, 25.99% of PAMs were apoptotic. When GZ201801-ASFV and ΔCD2v-ASFV were co-infected, 28.67% of PAMs were apoptotic (Figure 3).

### 3.4. Phospho-NF-κB p65 and p-IκB of ΔCD2v/ΔMGF360-505R-ASFV in Primary Swine Macrophages

Western blotting was used to measure the expression of phospho-NF-κB p65 protein and p-IκB protein levels at 3, 6, and 12 h in each group of PAMs. This result suggests that the level of phospho-NF-κB p65 and p-IκB protein increased after GZ201801-ASFV infection, and this effect was abrogated by CD2v and MGF360-505R deficiency. The level of phospho-NF-κB p65 and p-IκB protein after ΔMGF360-505R-ASFV infection is more than the GZ201801-ASFV infection, and the level of phospho-NF-κB p65 and p-IκB protein after ΔCD2v-ASFV infection is less than GZ201801-ASFV infection. Expression of Tubulin was used as a positive control (Figure 4). And the gray analysis of protein bands was carried out (Appendix A).

### 3.5. MGF360-12L, MGF360-13L, and MGF505-2R Inhibit NF-κB Promoter

The luciferase reporter gene internal reference plasmids pRL-TK and NF-κB-Luc reporter plasmids were mixed, and co-transfected with pCAGGs-MGF360-12L, pCAGGs-MGF360-13L, pCAGGs-MGF360-14L, pCAGGs-MGF505-1R, pCAGGs-MGF505-2R, and pCAGGs-MGF505-3R plasmids separately, and Luciferase Reporter Assays were performed. This result suggests that MGF360-12L, MGF360-13L, and MGF505-2R of ASFV MGF360-505R significantly inhibit the NF-κB promoter compared with the control empty vector (Figure 5).

### 3.6. ΔCD2v/ΔMGF360-505R-ASFV Inhibits the Expression of IL-1β mRNA

To study the changes of IL-1β after the PAMs were infected with GZ201801-ASFV or ΔCD2v/ΔMGF360-505R-ASFV, IL-1β mRNA expression was measured by real-time RT-PCR. The expression of IL-1β mRNA in the GZ201801-ASFV infection group at 3 h and 12 h increased with the time of infection. The expression of IL-1β mRNA in the ΔCD2v/ΔMGF360-505R-ASFV infection group was significantly lower than in the GZ201801-ASFV infection group, at 3 h and 12 h (Figure 6).

## 4. Discussion

ASFV infection of domestic pigs or wild boars can cause severe hemorrhagic diseases, and the mortality rate is as high as 100% [45]. Because ASFV and the host have a complex interaction mechanism, ASFV can evade the host immune system [4]. Some pathogens kill macrophages by inducing apoptosis, thereby reducing the phagocytic ability of phagocytes to pathogens [46]. Using a DNA homologous recombination technique [1], we constructed ΔCD2v/ΔMGF360-505R-ASFV viruses by deleting gene segments encoding seven different proteins, includingMGF505-1R, MGF505-2R, MGF505-3R, MGF360-12L, MGF360-13L, MGF360-14L, and CD2v, which have previously been observed to be important for the virulence of different ASFVs [1,47]. This study shows that GZ201801-ASFV has obvious CPE after being infected with PAMs, while ΔCD2v/ΔMGF360-505R-ASFV infection produces significantly lower CPE compared with GZ201801-ASFV. Through flow cytometry, it was found that the percentage of PAMs apoptosis caused by ΔCD2v/ΔMGF360-505R-ASFV infection was significantly lower than that of GZ201801-ASFV infection. When GZ201801-ASFV and ΔCD2v/ΔMGF360-505R-ASFV are co-infected, the CPE of PAMs and the percentage of PAMs cell apoptosis are also significantly lower than that of the GZ201801-ASFV infection.

Apoptotic macrophages can also cause the host’s inflammatory response by secreting pro-inflammatory cytokines [23]. In recent years, more and more studies have been conducted on ASFV infection in order to activate the host NF-κB signaling pathway. Although different viruses activate the upstream signaling molecules of the NF-κB pathway by different mechanisms, they will eventually converge the signal to the IκB kinase (IKK) complex. It activates IκB to cause a cascade of signals, and finally initiates the transcription of target genes [48]. Therefore, it shows that IKK, the IκB kinase complex, is a key node that activates the NF-κB signaling pathway caused by various external stimuli. The cytokines and chemokines (TNF-α and IL-1β) produced after the activation of the NF-κB signaling pathway can reactivate the NF-κB signaling pathway and cause a cascade reaction in the host [48]. Different signaling pathways respond to different levels of IL-1β, which can lead to host genotoxic damage and cell apoptosis, or affect cell growth. In the environment of high levels of IL-1β, cells that have undergone genotoxic damage in the host will participate in the apoptosis pathway [49]. This study shows that GZ201801-ASFV infection of PAMs activates the NF-κB signaling pathway and promotes the expression of IL-1β. However, the up-regulated expression levels of phospho-NF-κB p65 and p-IκB proteins and IL-1β by ΔCD2v/ΔMGF360-505R-ASFV infection was significantly lower than that of GZ201801-ASFV. When the MGF360-505 deletion strain is infected with PAMs, it activates the host’s NF-κB signaling pathway, while the CD2v-deficient strain inhibits the host’s NF-κB signaling pathway after infecting PAMs, indicating that MGF360-505 has the effect of inhibiting NF-κB and CD2v has the effect of activating NF-κB. We further found that MGF360-12L, MGF360-13L, and MGF-505-2R suppress the promoter activity of NF-κB, and CD2v activates the NF-κB signaling pathway.

The activation and regulation mechanism of apoptosis is an extremely complex process regulated by the body’s internal mechanisms, and it participates in a variety of life activities. In depth research on the molecular mechanism of ASFV infection and activation of cell apoptosis will help to develop and use cell apoptosis as a new weapon against ASFV replication and provide a theoretical basis for the prevention and treatment of ASF. In summary, our data demonstrated that ΔCD2v/ΔMGF360-505R-ASFV infection caused PAMs apoptosis and the expression levels of ASFV p30 protein are significantly lower than the GZ201801-ASFV infection. Also, ΔCD2v/ΔMGF360-505R-ASFV infection upregulates the expression levels of phospho-NF-κB p65 and p-IκB proteins and IL-1β production are significantly lower than the GZ201801-ASFV infection. Additionally, MGF360-12L, MGF360-13L, and MGF-505-2R suppress the promoter activity of NF-κB (Figure 7). These findings suggest that ΔCD2v/ΔMGF360-505R-ASFV reduces the apoptosis of PAMs by inhibiting the NF-κB signaling pathway and IL-1β, which might provide a theoretical basis for the development of an effective vaccine for ASFV.

## Figures and Tables

**Figure 1 vaccines-09-01371-f001:**
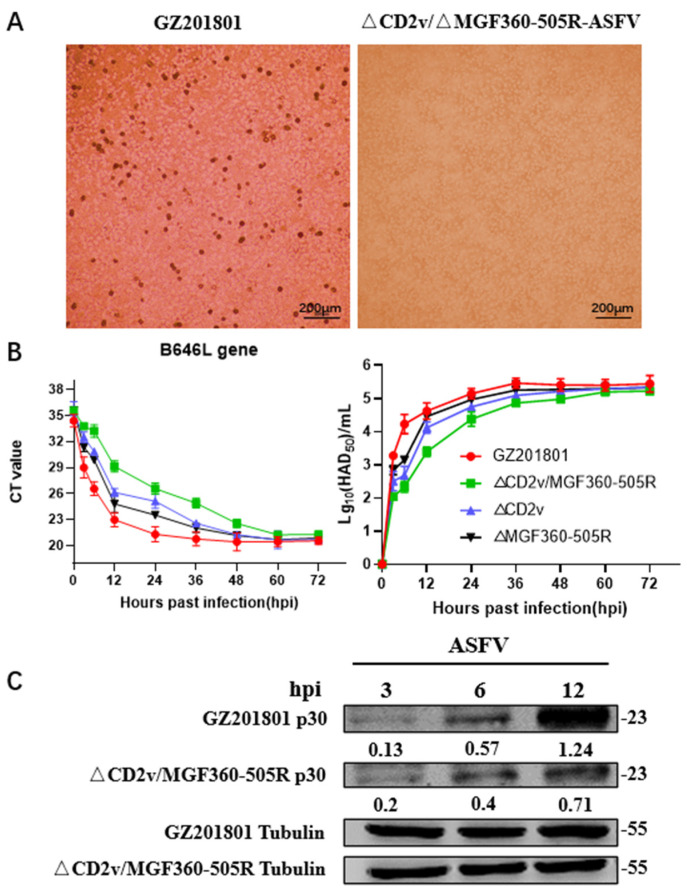
Detection of GZ201801-ASFV, ΔMGF360-505R-ASFV, ΔCD2v-ASFV, and ΔCD2v/ΔMGF360-505R-ASFV in primary PAMs. (**A**) HAD appeared in primary PAMs infected with GZ201801-ASFV and ΔCD2v/ΔMGF360-505R-ASFV. (**B**) The growth curves of PAMs infected with GZ201801-ASFV and ΔCD2v/ΔMGF360-505R-ASFV. (**C**) Western blotting was used to detect the expression of p30 protein. PAMs were either infected with GZ201801-ASFV, ΔMGF360-505R-ASFV, ΔCD2v-ASFV, and ΔCD2v/ΔMGF360-505R-ASFV at a MOI of 0.1.

**Figure 2 vaccines-09-01371-f002:**
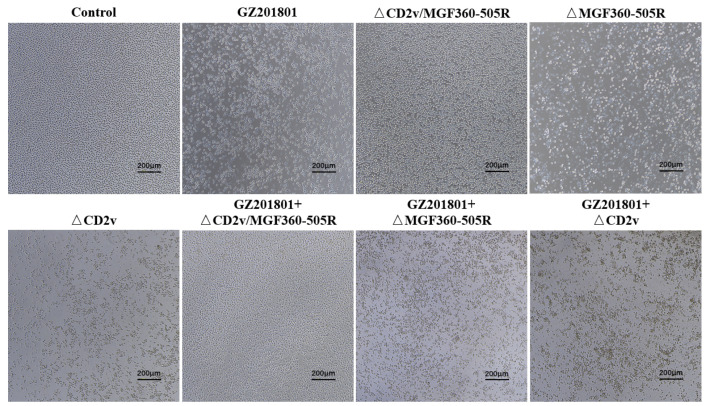
The CPE of infected primary PAMs with GZ201801-ASFV, ΔCD2v/ΔMGF360-505R-ASFV, ΔMGF360-505R-ASFV, and ΔCD2v-ASFV. GZ201801-ASFV infection of PAMs produces more serious CPE. ΔCD2v/ΔMGF360-505R-ASFV or co-infection with GZ201801-ASFV and GZ201801-ASFV produced less CPE in PAMs. PAMs were either infected with GZ201801-ASFV, ΔCD2v/ΔMGF360-505R-ASFV, ΔMGF360-505R-ASFV, or ΔCD2v-ASFV at a MOI of 0.1.

**Figure 3 vaccines-09-01371-f003:**
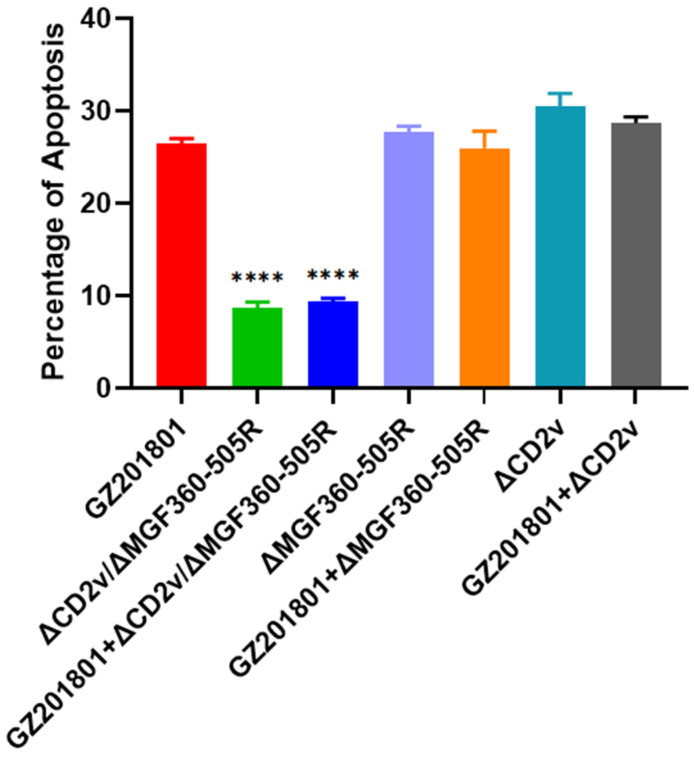
The apoptosis of GZ201801-ASFV, ΔCD2v/ΔMGF360-505R-ASFV, ΔMGF360-505R-ASFV, and ΔCD2v-ASFV are infected in primary PAMs. GZ201801-ASFV caused 26.49% of cell apoptosis after infection with PAMs, ΔCD2v/ΔMGF360-505R-ASFV caused 8.72% of cell apoptosis after infection with PAMs, ΔMGF360-505R-ASFV caused 27.67% of cell apoptosis after infection with PAMs, ΔCD2v caused 30.45% of cell apoptosis after infection with PAMs, GZ201801-ASFV and ΔCD2v/ΔMGF360-505R-ASFV caused 9.35% of cell apoptosis after co-infection with PAMs, GZ201801-ASFV and ΔMGF360-505R-ASFV caused 27.67% of cell apoptosis after co-infection with PAMs, and GZ201801-ASFV and ΔCD2v caused 30.45% of cell apoptosis after co-infection with PAMs. PAMs were either infected with GZ201801-ASFV, ΔCD2v/ΔMGF360-505R-ASFV, ΔMGF360-505R-ASFV, or ΔCD2v-ASFV at a MOI of 0.1. Significant differences compared with the control group are denoted by **** (*p* < 0.001).

**Figure 4 vaccines-09-01371-f004:**
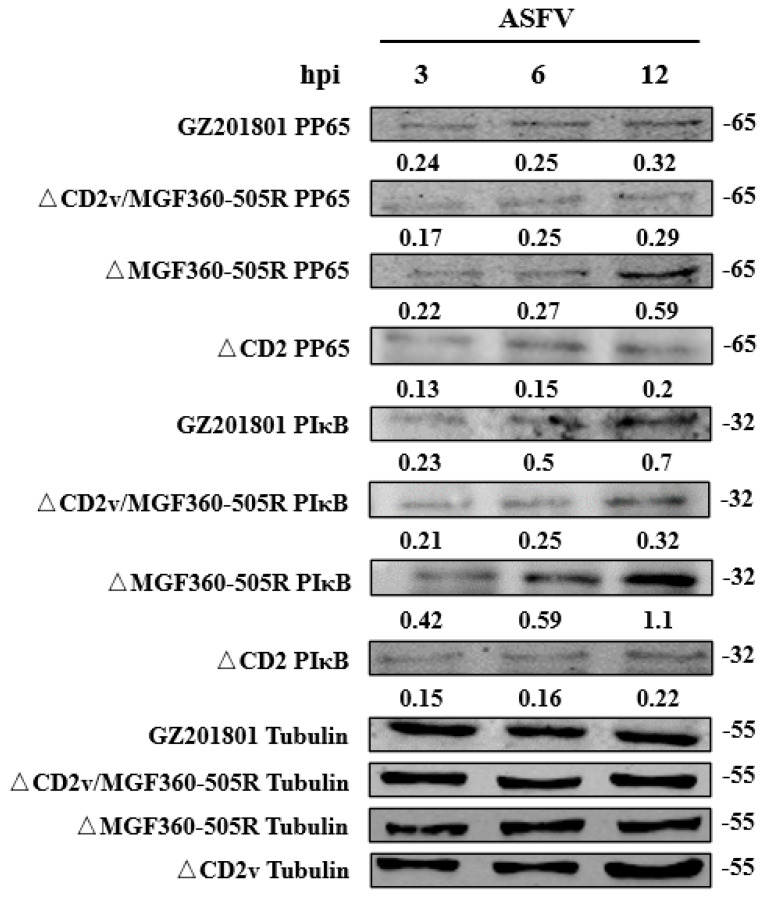
NF-κB signaling pathway analysis after ASFV infection. Western blotting was used to measure the expression of PP65 and PIκB protein at 3, 6, and 12 h in each group of GZ201801-ASFV-, ΔMGF360-505R-ASFV-, ΔCD2v-ASFV-, and ΔCD2v/ΔMGF360-505R-ASFV-infected PAMs. Expression of tubulin was used as a positive control. PAMs were either infected with GZ201801-ASFV or ΔCD2v/ΔMGF360-505R-ASFV at a MOI of 0.1.

**Figure 5 vaccines-09-01371-f005:**
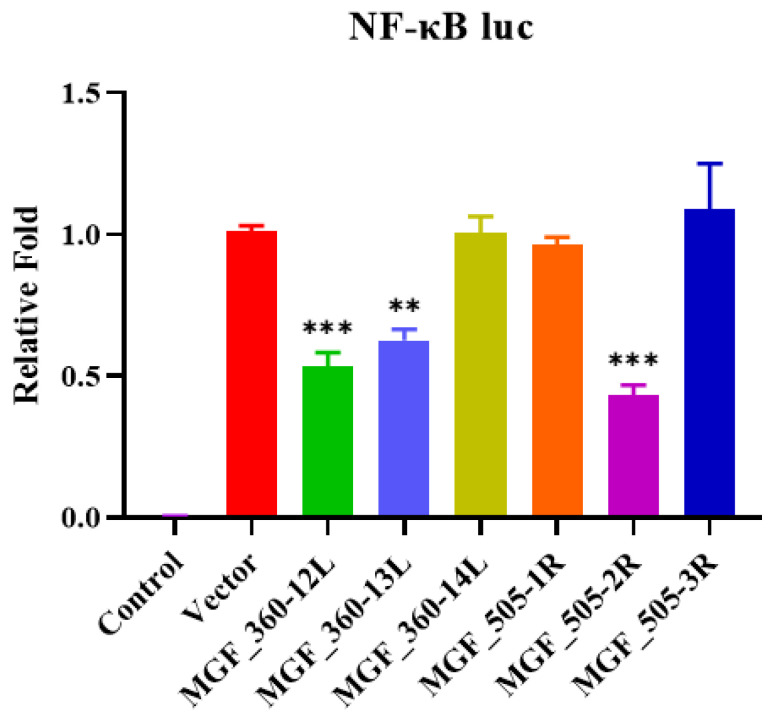
MGF360-12L, MGF360-13L, and MGF505-2R interfere with the NF-κB signaling pathway. Plasmid encoding MGF360-12L, MGF360-13L, MGF360-14L, MGF505-1R, MGF505-2R, MGF505-3R, or empty vectors were co-transfected into PK-15 cells with the NF-κB-Luc promoter reporter plasmid and with pRL-TK. After 24 h, luciferase activity was determined by dual-luciferase assay. Each datum represents results of three independent experiments (means ± SD). Significant differences compared with the control group are denoted by ** (*p* < 0.01) and *** (*p* < 0.001).

**Figure 6 vaccines-09-01371-f006:**
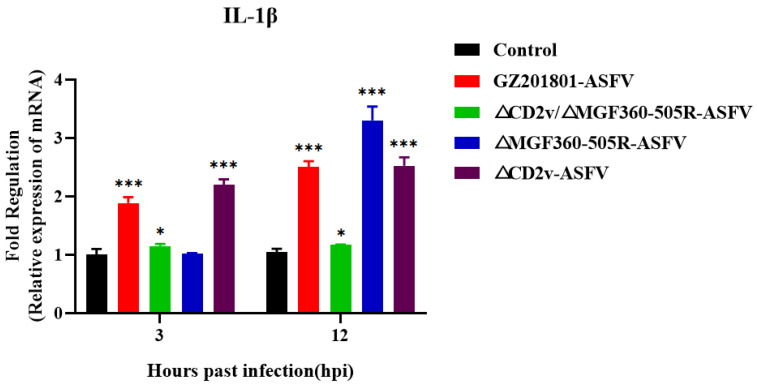
Changes in expression of IL-1β mRNA. Real-time RT-PCR was used to assess the mRNA expression of IL-1β in PAMs incubated with GZ201801-ASFV, ΔCD2v/ΔMGF360-505R-ASFV ΔCD2v, andΔMGF360-505R-ASFV after 0, 3, 6, and 12 h. Increased expression of IL-1β mRNA was seen in GZ201801-ASFV-infected cells and this effect was abrogated by CD2v and MGF360-505R deficiency. Each datum represents results of three independent experiments (means ± SD). Significant differences compared with the control group are denoted by * (*p* < 0.05) and *** (*p* < 0.001). PAMs were either infected with GZ201801-ASFV or ΔCD2v/ΔMGF360-505R-ASFV at a MOI of 0.1.

**Figure 7 vaccines-09-01371-f007:**
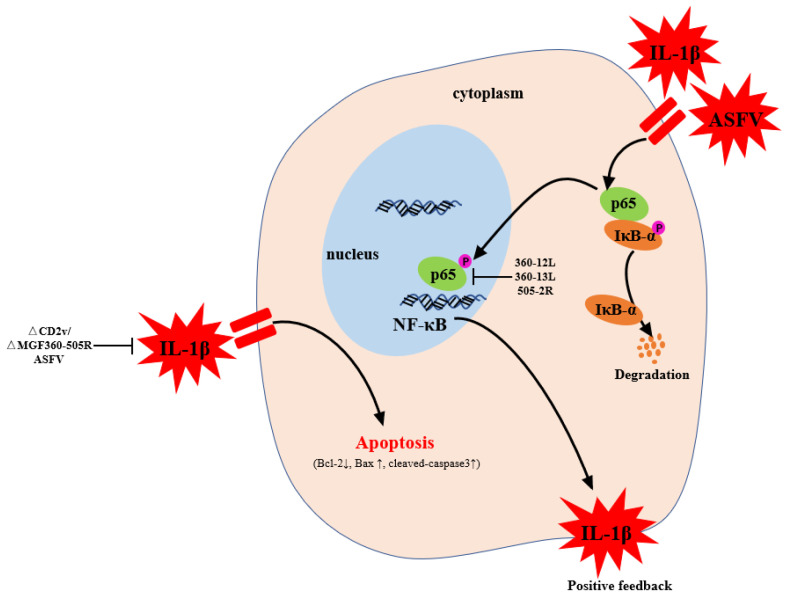
The mechanism of ΔCD2v/ΔMGF360-505R-ASFV inhibits apoptosis after infecting PAMs cells. After infection with PAMs, GZ201801-ASFV promotes the release of IL-1β to induce cell apoptosis. After ΔCD2v/ΔMGF360-505R-ASFV infection with PAMs, the activation of NF-κB pathway and IL-1β are significantly lower than GZ201801-ASFV. Therefore, the percentage of apoptosis induced by ΔCD2v/ΔMGF360-505R-ASFV infecting PAMs was significantly lower than in GZ201801-ASFV.

**Table 1 vaccines-09-01371-t001:** Primer sequences were used in this study for PCR and real-time PCR in pigs.

Gene	Primer Sequence (5′–3′)
CADC-B646L-rPCRF	ATAGAGATACAGCTCTTCCAG
CADC-B646L-rPCRR	GTATGTAAGAGCTGCAGAAC
CADC-B646L-Probe	FAM-TATCGATAAGATTGAT-MGB
B646L-F	TGAAATAAAATGGAAGCCCACAGATC
B646L-R	ACACTGTACAACATTGCGTAAAAGC
GAPDH-F	GCAAAGACTGAACCCACTAATT
GAPDH-R	TTGCCTCTGTTGTTACTTGGAG
IL-1β-F	ACCTGGACCTTGGTTCTCTG
IL-1β-R	CATCTGCCTGATGCTCTTG

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
