# Peer review of "The African Swine Fever Virus with MGF360 and MGF505 Deleted Reduces the Apoptosis of Porcine Alveolar Macrophages by Inhibiting the NF-κB Signaling Pathway and Interleukin-1β"

_vaccines, 2021, doi:10.3390/vaccines9111371_

Round 1

Reviewer 1 Report

This paper can be improved by the comparison of ΔMGF360-505R-ASFV to determine the roles of the CD2v and the MGF360-505R.

Line 121 remove the underline and check the font size.

Line 199 Why was the replication of ΔMGF360-505R-ASFV not done? This is important to know the role of CD2v.

Likewise for the cytopathic effect why was ΔMGF360-505R-ASFV not used and also used for co-infection?

Same issue with the apoptosis studies why was  ΔMGF360-505R-ASFV not used? Likewise for the IL-1B.

Line 232 replace with "The CPE of infected primary PAMs with GZ201801-ASFV  and ΔCD2v/ΔMGF360-505R-ASFV. GZ201801-ASFV infection of PAMs produces more serious CPE. ΔCD2v/ΔMGF360-505R- 233 ASFV or co-infection with GZ201801-ASFV and 201801-ASFV produced less CPE in PAMs.

Author Response

  1. The underscore on line 121 has been removed, and the word size has been changed.
  2. Our laboratory has conducted research on the replication of ΔMGF360-505R-ASFV, but since this article mainly studies ΔCD2v/ΔMGF360-505R-ASFV, the results of ΔMGF360-505R-ASFV are not included in the article.
  3. Our laboratory has studied the cytopathic effect of ΔMGF360-505R-ASFV. At the same time, it has also conducted co-infection studies on ΔMGF360-505R-ASFV and the wild strain ASFV. However, due to the ΔMGF360-505R-ASFV and the wild strain ASFV Co-infection cannot reduce the cytopathic effect, and this article mainly studies ΔCD2v/ΔMGF360-505R-ASFV, so the results of ΔMGF360-505R-ASFV are not included in the article.
  4. Our laboratory has studied the apoptosis and IL-1β induced by ΔMGF360-505R-ASFV infection. However, due to ΔMGF360-505R-ASFV infection, the induced apoptosis and IL-1β cannot be reduced, and this article mainly studies ΔCD2v/ΔMGF360-505R-ASFV, so the results of ΔMGF360-505R-ASFV are not included in the article.
  5. Line 232 has been modified.

Reviewer 2 Report

It is unclear what virus is being used see point #1 below. But assuming from the title that this is the 7 gene  MGF deletion that is published by others and not a single deletion of MGF360-505R. If it is the single deletion, the paper needs to be rewritten to reflict only that gene. There are multiple papers deleting individual genes in this area or gene families.  They have also been linked by others to have an effect on the immune response pathway PMID: 27043071, 34517008, 33712518.  There is also no work done in the context of the virus, to show the differential effect of each individual gene.  The results are expected based on the previous results and don’t build upon them or really add much information.

  1. It is not clear if the deleted virus contains only one gene deleted or both MGF 360 and MGF505 genes deleted ( if so how many and which ones are deleted) . A reference to this recombinant virus, or details on how it is made is required.
  2. References are missing on other gene deletions in this MGF area by other groups and should be included in the discussion.
  3. Figure 2 and 3 should be combined. Rather than show figure 3 as is it should have the % in a bar graph, along with different trials. It is also possible that this result is due to a longer replication cycle. Different time points need to be used to see if the MGF deleted and CD2MGF catch up to the WT rate of apoptosis. For example do the PAM cells survive 72hrs after the WT cells have all undergone CPE. This would be very interesting if it is the case.
  4. Figure 5 how do you know the cells were equally transfected? And that the proteins are being expressed at similar levels to make the conclusions drawn from this figure.
  5. MOIs need to be added to all figure legends.
  6. Using the expression plasmids and your deletion virus, can you restore the wt phenotype?
  7. There are differences between the CD2+MGF deleted virus and the MGF deleted virus, however both are not used in all of the Experiments, and the role of CD2 is ignored. Figure 6 needs to have the recombinant virus with only the MGF deletions, Figure 5 lacks CD2
  8. Figure one should show titrations of the growth curve, and it should be extended to at least 48hrs, 72 if it doesn’t flatten out. This is necessary to see if virus is produced in both recombinant viruses.

Author Response

  1. Line 94 of this article clarifies the details of the missing genes in ΔCD2v/ΔMGF360-505R-ASFV. The deletion strains used in this article are all modified by our laboratory, and the relevant content will be published in other articles.
  2. References related to missing genes have been added to the discussion section.
  3. In this experiment, after observing the cytopathic phenomenon in Fig. 2, cell apoptosis was further detected, and the result in Fig. 3 was obtained. This article aims to study the early inhibitory effect of ΔCD2v/ΔMGF360-505R-ASFV infection on PAM cell apoptosis, so only early cell apoptosis is detected.
  4. The experimental methods and principles of this part refer to the pMGF505-7R published on Plos Pathogens by Researcher Weng Changjiang from Harbin Veterinary Research Institute to determine the pathogenicity of African swine fever virus infection by inhibiting the production of IL-1β and type I interferon The article, Chinese Academy of Agricultural Sciences. The article has been uploaded.
  5. MOI has been added to all legends.
  6. Because the susceptible cell of ASFV is PAM, but the transfection efficiency of PAM is very low, only 1%-5%, so the deletion virus and expression plasmid are not used for phenotypic recovery experiments.
  7. This article mainly studies ΔCD2v/ΔMGF360-505R-ASFV. When observing the cytopathic effect, only ΔCD2v/ΔMGF360-505R-ASFV can inhibit the cytopathic effect, so only relevant research results are shown. In Figure 4, the PP65 protein expression level of PAM cells infected with ΔCD2v/ΔMGF360-505R-ASFV and ΔMGF360-505R-ASFV has been detected. The results showed that the level of PP65 protein in PAM cells infected with ΔCD2v/ΔMGF360-505R-ASFV was lower than that of WT, and the expression level of PP65 protein in PAM cells infected with ΔMGF360-505R-ASFV was higher than that of WT, indicating that CD2v activated NF-κB , so CD2v is not detected in Figure 5.
  8. The virus infection time was extended to 72h.

Round 2

Reviewer 1 Report

The authors did not respond sufficiently to the reviewers comments and did not include the requested data in the paper to demonstrate the role of CDV2 in the MTF deleted virus.

Author Response

Thank you for your review. The relevant experimental data of the strain that lacks CD2v has been added to the article, and it has been marked in yellow. Please check it.

Reviewer 2 Report

The reviewers didn’t answer questions from my previous review. Replying that “the relevant content will be published in other articles is unacceptable”  It appears that this paper, is remaking the same gene deletions as has been reported by other groups.  It is not referenced in this way and is written as if this gene deletion is new. In addition the conclusions can not currently be interpreted from the results.  The authors do not try to answer several questions, but make excuses as to why they can’t do the experimetns.

  1. It should specifically state, as described in ref 1 we made the same gene deletion in the MGF region. And as described in (PMID: 32124180) where an addition to the MGF gene deletion was a deletion in CD2.  The differences between these two viruses and the one in this manuscript should be given. There is nothing wrong with making the same mutants, but they need to be referenced correctly.  The methodology to create this in house copy should be described so that it could be repeated by other groups.

  1. There are differences between the CD2+MGF deleted virus and the MGF deleted virus, however both are not used in all of the Experiments, and the role of CD2 is ignored. Figure 6 needs to have the recombinant virus with only the MGF deletions, Figure 5 lacks CD2.  This data is required to be able to make the conclusions in the paper.

  1. This was not answered from my previous round of review: Figure 5 how do you know the cells were equally transfected? And that the proteins are being expressed at similar levels to make the conclusions drawn from this figure.
  1. The sequence of the recombinant virus should the included in supplementary data. Along with information about the number of aligned NGS reads.
  2. Scheme 2, there is different exposures to the fluorescence images.

Author Response

Thank you for your review. The relevant experimental data of the strain that lacks CD2v has been added to the article, and it has been marked in yellow. Please check it.

1. The article has been supplemented with relevant experimental data about the deletion of CD2v virus alone, and has been marked in yellow.

2. The sequence of the missing virus has been uploaded, please check it.

3. Regarding the expression level of the transfection plasmid, in this experiment, we transfected the same amount of plasmid into PK-15 cells and tested the luciferase reporter gene. This method is based on the Harbin Institute of Veterinary Research, Chinese Academy of Agricultural Sciences Experiments conducted on published articles. By comparing the expression of NF-κB luciferase in the control group and the experimental group transfected with viral plasmids, we judged whether the viral protein can inhibit the transcription level of NF-κB, and obtained the corresponding results. Our laboratory did not detect CD2v's detection of NF-κB-luc plasmid expression, so the result is not shown in Figure 5. So we added a virus experiment that deleted CD2v alone, and got that CD2v can activate NF-κB.

Round 3

Reviewer 1 Report

Figure 3 the side color legend is redundant and can be removed as the specific virus treatment is on the graph axis.

Line 348 please fix the sentence structure.

Author Response

Thank you for your comments. I have proposed amendments to the comments, please check.

1. The legend on the right side of Figure 3 has been removed.

2. Line 348 in the article has been revised, and "It" has been removed.

Reviewer 2 Report

Line 317, reference 43 does not show a recombination technique I believe this should be reference 1.

Line 320, same comment reference 43 does not reference these genes I believe this should be reference 1.

Author Response

Thank you for your comments. I have proposed amendments to the comments, please check.

  1. Reference 43 in line 317 has been revised to reference 1.
  2. Reference 43 in line 320 has been revised to reference 1.
  3. Reference 43 has been deleted. And the order of all references has been adjusted correctly.

This manuscript is a resubmission of an earlier submission. The following is a list of the peer review reports and author responses from that submission.